# Prenatal Hyperglycemia Exposure and Cellular Stress, a Sugar-Coated View of Early Programming of Metabolic Diseases

**DOI:** 10.3390/biom10101359

**Published:** 2020-09-23

**Authors:** Jessica Tozour, Francine Hughes, Arnaud Carrier, Didier Vieau, Fabien Delahaye

**Affiliations:** 1Department of Obstetrics and Gynecology, NYU Winthrop Hospital, Mineola, NY 11501, USA; Jessica.tozour@nyulangone.org; 2Obstetrics & Gynecology and Women’s Health, Division of Maternal-Fetal Medicine, Montefiore Medical Center/Albert Einstein College of Medicine, Bronx, NY 10461, USA; fhughes@montefiore.org; 3Institut Pasteur de Lille, U1283-UMR 8199 EGID, Université de Lille, Inserm, CNRS, CHU Lille, F-59000 Lille, France;arnaud.carrier@pasteur-lille.fr; 4BiologyDepartment, LilNCog Lille Neurosciences and Cognition U 1172, Université de Lille, Inserm, CHU Lille, F-59000 Lille, France;didier.vieau@univ-lille.fr

**Keywords:** programming, metabolic disease, prenatal hyperglycemia, oxidative stress

## Abstract

Worldwide, the number of people with diabetes has quadrupled since 1980 reaching 422 million in 2014 (World Health Organization). This distressing rise in diabetes also affects pregnant women and thus, in regard to early programming of adult diseases, creates a vicious cycle of metabolic dysfunction passed from one generation to another. Metabolic diseases are complex and caused by the interplay between genetic and environmental factors. High-glucose exposure during in utero development, as observed with gestational diabetes mellitus (GDM), is an established risk factor for metabolic diseases. Despite intense efforts to better understand this phenomenon of early memory little is known about the molecular mechanisms associating early exposure to long-term diseases risk. However, evidence promotes glucose associated oxidative stress as one of the molecular mechanisms able to influence susceptibility to metabolic diseases. Thus, we decided here to further explore the relationship between early glucose exposure and cellular stress in the context of early development, and focus on the concept of glycemic memory, its consequences, and sexual dimorphic and epigenetic aspects.

## 1. Introduction

The late 1980s marked a fundamental period in epidemiology with the observation by Barker and colleagues of a positive correlation between infant mortality rates and mortality from ischemic heart disease, bronchitis and stomach cancer in adulthood [1]. With the studies following Barker’s 1986 publication, low birth weight became a symbol of poor fetal growth due to inadequate intrauterine conditions associated with increased risk for metabolic and age-related diseases. Later evidence suggests that programming of metabolic diseases is not limited to food restriction but also found in cases of over nutrition. This has been described as a U-shaped association of birth weight and adult disease risk [2]. This first observation was then confirmed by numerous studies in the general population [3,4] leading to the conclusion that both extremes of fetal growth have a measurable influence on long-term health creating a vicious cycle of metabolic dysfunction passed from one generation to another. Such a phenomenon has been the focus of many studies in the past years as the underlying mechanisms have yet to be elucidated.

Oxidative stress, a key factor in cellular stress, has been identified in obesity, insulin resistance and diabetes mellitus [5] suggesting a link between cellular stress and metabolic disease susceptibility. Oxidative stress can be initiated after high-glucose exposure in various tissues [6]. Our group has demonstrated that the long-term consequences of early exposure could be mediated through high glucose related impaired response to oxidative damage [7] in mesenchymal stem cell. These observations suggest that oxidative stress influences and is influenced by cellular metabolic status. Thus, oxidative stress could be a mediator in the relationship between high glucose exposure and susceptibility to metabolic diseases. We propose to further review this relationship between glucose exposure and cellular stress in the context of age-related and metabolic disease early programming.

## 2. Hyperglycemia and Glucose Metabolism

The primary utility of glucose in the cell is to generate energy, which begins with glycolysis resulting in pyruvate and acetyl-CoA, and further processed through the tricarboxylic acid cycle (TCA) or Krebs cycle to provide nicotinamide adenine dinucleotide (NADH) for adenosine triphosphate (ATP) production via oxidative phosphorylation. Alternative pathways can be engaged by glucose and glycolytic intermediates when glucose is in abundance in the cell or to compensate the inhibition of specific glycolytic enzymes. Many of these pathways are strongly related to reactive oxygen species (ROS) production or elimination, thus high-glucose exposure will often result in reactive oxygen species accumulation (Figure 1). Such phenomenon referred to as glucotoxicity provides a plausible mechanism recapitulating the association between high-glucose exposure and increased metabolic disease sensitivity through cellular stress related mechanisms.

The polyol pathway is considered a nutrient-sensing pathway increasingly relevant as intracellular glucose rises. This pathway generates fructose by converting glucose to sorbitol catalyzed by aldose reductase followed by sorbitol dehydrogenase mediated oxidation [8]. During this process NADH is generated while nicotinamide adenine dinucleotide phosphate (NADPH) is consumed. NADPH is required for the production of the reduced form of glutathione (GSH) an important non-enzymatic antioxidant [9]. If glucose levels in the cell are high enough, the polyol pathway will outcompete the GSH synthesis pathway leading to a decrease in antioxidant capacity in the cell.

The hexosamine biosynthetic pathway (HBP) is another important nutrient sensing pathway that shunts in more glucose under a hyperglycemic state. The rate-limiting enzyme of this pathway glutamine:fructose-6-phosphate amdiotransferase (GFAT) converts fructose-6-phosphate to N-acetylglucosamine-6-phosphate, which either gets phosphorylated again to become N-acetylglucosamine-1,6-phosphate or further process to uridine-diphosphate N-acetylglucosamine (UDP-GlcNAc). UDP-GlcNAc can post-translationally modify proteins through the action of O-GlcNAc transferase. O-glycosylation of proteins can modify their function and has been documented in some high glucose and diabetic models. Patti et al. (1999) found that the insulin receptor substrates-1 and -2 (IRS-1, IRS-2) became O-GlcNacylated after hexosamine pathway activation and that there was reduction in IRS-1 mediated insulin signaling [10]. Apart from the protein O-GlcNAc modification, activation of HBP produces H_2_O_2_, and may independently induce cellular dysfunction in pancreatic β-cells [6]. This pathway was also implicated in interfering with the required influx of Ca^+2^ into β-cells that is required for insulin secretion [11].

Increased glucose flux also allows for the intermediates fructose-6-phosphate and glyceraldehyde-3-phosphate to be converted to diacylglycerol (DAG), an important activating cofactor of protein kinase-C (PKC), -β, -δ, -α [9]. PKC can activate many signaling processes within the cell, as well as lead to gene expression changes in a tissue specific manner. PKC can also activate NOX, a prominent superoxide generating protein in the cells [12]. Additionally, glyceraldehyde-3-phosphate can undergo autoxidation, forming a by-product called enediol [13]. During this autoxidation process, hydrogen peroxide is produced as well as alpha-ketoaldehyde which can contribute to protein modifications [14].

Advanced glycation end products (AGEs) are another consequence of increased intracellular high glucose. Similar to that described for O-GlcNAc modification to proteins, glycation of proteins can also affect their function. In addition, glycated proteins have been demonstrated to increase reactive oxygen species through PKC-mediated NOX2 activation [15].

In physiological instances, mitochondrial oxidative phosphorylation produces the majority of cellular reactive species. When glucose is increased in the cell, more glucose becomes processed and oxidized through the TCA cycle, generating a large amount of NADH and flavin adenine dinucleotide (FADH_2_) for the electron transport chain leading to a critical rise in electron saturating complex III [9]. When this occurs, coenzyme Q begins donating electrons to molecular oxygen, creating the formation of superoxide radicals. The increased production of NADH through the polyol pathway can have an additive effect on this mitochondrial dysfunction [16]. Therefore, the increase in glucose metabolism plays a major role in ROS production.

Finally, high glucose conditions are also associated with increased flux into the afore-mentioned alternative pathways through the inhibition of glyceraldehyde-3-phosphate dehydrogenase (GAPDH). High glucose conditions promote the polyADP-ribosylation of GAPDH which decreases its activity [17]. This modification of GAPDH is mediated by poly(ADP-ribose) polymerase (PARP), an enzyme that becomes activated in response to DNA strand breaks, which was also demonstrated under high glucose conditions. Therefore, when high glucose and ROS levels exceed the protective mechanisms of the cell, inactivation of GAPDH causes an increased flux of glucose into the alternative pathways leading to further increased ROS levels in the cell.

## 3. Hyperglycemia, Oxidative Stress and Diabetes Susceptibility

Reactive oxygen species have been proposed as mediators for insulin resistance after high-glucose exposure as illustrated in different key metabolic tissues. ROS may participate in the development of type 2 diabetes through increasing pancreatic β-cell dysfunction, that occurs when ROS levels exceed the cell’s already low antioxidant defenses [18]. Indeed, chronic high-glucose exposure of rat islets increased hydrogen peroxide production [6]. This increased ROS leads to decreased activity of pancreatic and duodenal homeobox-1 (PDX-1), an important transcription factor for β-cell function, including insulin biosynthesis and secretion [13]. ROS mediates the activation of JNK pathway which causes the nucleocytoplasmic translocation of PDX-1 leading to decreased DNA-binding of the transcription factor [19]. Therefore, ROS and JNK pathway activation are likely causes for the β-cell dysfunction in type 2 diabetes [20]. Furthermore, high-glucose related oxidative stress was evident in pancreatic β-cells with the detection of 8-oxodG, a marker of DNA damage due to oxidative stress [21], thus, high-glucose related DNA damage could further contribute to the decline in β-cell function.

Oxidative stress induced by chronic high glucose is also involved in the development of insulin resistance in adipose and muscle tissues. High glucose exposure in these tissues induced increased ROS, which led to the down regulation and decreased localization of GLUT4 to the cell surface in response to insulin [22]. Therefore, high glucose induces insulin resistance that decreases glucose uptake in response to insulin. High glucose environment and increased free fatty acids can lead to insulin resistance in the liver, thus interfering with gluconeogenesis [23]. In an insulin resistance state, hepatic glucose output remains high, further contributing to hyperglycemia.

High glucose-induced oxidative stress plays an important role in endothelial cell dysfunction leading to the microvascular and macrovascular complications associated with type 2 diabetes. Oxidative stress in microvascular endothelial cells can lead to the development of diabetic retinopathy, dementia, nephropathy and neuropathy [24].

## 4. Hyperglycemia and Early Programming of Metabolic Diseases

### 4.1. Maternal Glucose Metabolism during Pregnancy

The developing fetus creates an increasing nutrient demand on maternal metabolism and homeostasis. Therefore, during pregnancy, changes occur to maternal glucose to meet these increased nutrient demands [25,26,27,28].

During normal pregnancy, the first trimester retains normal glucose tolerance, insulin sensitivity and hepatic glucose production [27,28,29]. However, monitoring of insulin sensitivity and glucose tolerance through the length of pregnancy reveals a progressive increase in insulin resistance of peripheral tissues responsible for a decrease in glucose disposal by muscle and adipose tissue that divert more glucose to the fetal-placental unit [27,28,29]. By late pregnancy, insulin sensitivity is decreased to more than half of normal levels [26]. The liver is responsible for glucose production in the body through gluconeogenesis and this process is regulated in response to insulin and glucose levels [30]. During pregnancy, gluconeogenesis increases up to 30% to accommodate the increased nutrient demand [29,31]. Furthermore, post-prandial glucose uptake by the liver is suppressed during pregnancy and glycogen synthesis is reduced [32]. Therefore, the changes in both peripheral tissues and liver glucose production increase glucose availability to the fetus (Figure 2). As glucose gets prioritized for fetal utilization through reduced uptake by maternal tissues, lipolysis increases during pregnancy allowing for fatty acids to be used as an energy fuel for the mother [33]. To compensate for the increased insulin resistance, both basal and post-prandial insulin secretion increase as pregnancy progresses [34,35]. The combination of decreased insulin sensitivity and increased insulin secretion maintains relatively normal glucose levels in late pregnancy [26].

The cellular mechanisms responsible for the decrease in peripheral insulin sensitivity as well as increased insulin secretion are only starting to become elucidated. Factors produced by the placenta such as human placental lactogen, placental growth hormone, and TNF-α progressively rise during pregnancy to induce peripheral insulin resistance [36,37] through disruption of intracellular insulin signaling [25,38,39]. Adiponectin, a protein released from adipose tissue, increases insulin sensitivity in skeletal muscle and liver [40]. During pregnancy, adiponectin secretion is decreased from maternal adipose tissue influencing maternal insulin resistance [41]. Intrinsic factors affecting skeletal muscle insulin signaling have also been associated with pregnancy. In late pregnancy, insulin receptor tyrosine kinase activity was inhibited compared to non-pregnant controls due to increased phosphorylation of the inactivating serine/threonine residues [42]. In addition, insulin receptor substrate 1 (IRS-1) content has been found to be decreased in skeletal muscles with pregnancy [43]. In adipose tissue, the glucose transporter, GLUT4, is downregulated during pregnancy, contributing to the lack of insulin-induced glucose uptake by this tissue [44]. Transcription factor peroxisome proliferator-activated receptor (PPAR)-γ1, is an important regulator of adipose differentiation and insulin sensitivity through its activation of insulin sensitivity promoting factors like adiponectin [45]. During pregnancy, Catalano et al. (2002) found that PPARγ1 mRNA and protein expression was reduced by at least half [46]. This also supports the observation of reduced circulating adiponectin that acts on skeletal muscle and liver insulin sensitivity.

Interestingly, many of these alterations in cellular insulin sensitivity return to normal in the postpartum state [25], however if any of them become exaggerated, gestational diabetes mellitus (GDM) develops.

### 4.2. Hyperglycemia and Pregnancy

Diabetes during pregnancy can be described in one of three ways, type 1 diabetes mellitus (T1DM), type 2 diabetes mellitus (T2DM), and GDM. Type 1 and type 2 diabetes are considered pre-gestational diabetes, as they typically exist prior to conception. However, in contrast GDM is not present until after conception and does not develop until middle to late pregnancy in most cases [33].

Type 1 diabetes is an autoimmune disorder characterized by autoantibodies targeting pancreatic β-cells [47]. These autoantibodies recruit a variety of immune response cells, including CD8+ lymphocytes and macrophages that lead to the destruction of the insulin producing β-cells in the pancreatic islets. This eventually leads to a state of insulin depletion and hyperglycemia by young adulthood [47].

Type 2 diabetes is a complex and multifaceted disease that depends on the interplay between genetic predisposition and environment. The risk of developing type 2 diabetes increases with increasing body mass index (BMI), indicating that lifestyle and diet can play a role in its development. Ultimately, insulin resistance and hyperglycemia characterize this disease. Peripheral skeletal muscle and adipose tissue as well as liver fail to respond to insulin-stimulation to take up and stop producing glucose respectively [48]. In addition, pancreatic β-cell mass can become reduced, further limiting its production and secretion. Without the response to insulin, glucose is not efficiently cleared from the circulating plasma and hyperglycemia arises.

GDM is known as “glucose intolerance” with onset first recognized during pregnancy [49] and has become a growing concern in the U.S. with prevalence up to 16% [50]. While the pathogenesis of GDM is not yet fully understood, like type 2 diabetes it is complex and likely involves both genetic and environmental factors [51]. However, contrary to the dichotomous diagnosis of GDM, some of the elucidated mechanisms imply that insulin resistance and glucose intolerance in pregnancy are on a spectrum, with GDM on the extreme end [26,43]. As discussed above, pregnancy is associated with many changes in skeletal muscle, adipose tissue, liver, and pancreatic β-cells that result in insulin resistance and hyperglycemia. These factors are exaggerated when tested in GDM patients, making the glycemic levels much higher than what is considered physiologically normal. For instance, the pregnancy-related decreases in GLUT4 glucose-stimulated translocation, insulin receptor activity and insulin related substrate (IRS-1) expression are accentuated in muscle and adipose tissues of GDM mothers [42,43,46,52]. Furthermore, in GDM, the effects of pregnancy on the liver are accentuated with more glucose production, decreased hepatic glucose uptake, and glycogen storage [32], furthering the glucose load in circulation.

For women with pre-gestational diabetes, periconceptional glucose control is very important and correlates to risk of miscarriage. Before insulin therapy, when glucose levels were difficult to tightly control, infertility and neonatal mortality were common [47]. Even with insulin administration, if glucose levels are not optimal during organogenesis, the risk of diabetic embryopathy and congenital malformations are increased. Congenital malformations occur in 6–12% of pregnancies complicated by pre-gestational diabetes [53], and up to 5% in gestational diabetic pregnancies [54].

GDM and T2DM pregnancies that go to term, or are not complicated by malformation, have a variety of other risks for mother and baby, including macrosomia. Under the ‘Pederson hypothesis’ during diabetic pregnancy, it is posited that once the fetal pancreas has developed during mid-gestation, it secretes high levels of insulin in response to the high glucose levels that are crossing the placenta [55]. Maternal insulin does not readily cross the placenta, unless forced under high perfusion [56], but glucose is transported by facilitated diffusion through glucose transporters 1 and 3 [57]. By 10–15 weeks, the endocrine pancreas produces and secretes insulin, and in response to hyperglycemic conditions creates a hyper-insulinemic state [55,58]. The trophic effects of insulin stimulate fat and protein production, which often leads to fetal macrosomia [59,60].

Also, obesity commonly complicates diabetic pregnancies, further heightening the risk of excessive fetal growth [59]. Additional maternal risks, associated with a diabetic pregnancy, are increased risk of pre-eclampsia [61] and higher probability of developing type 2 diabetes 5–15 years after having a diabetic pregnancy [62].

## 5. Glycemic Memory and Long-term Consequences

Pregnancies affected by hyperglycemia, not only result in immediate complications to the fetus, but also have long-term effects on the infant’s adult health. Since diabetes during pregnancy is a major risk factor for high birth weight, many of the associations of large for gestational age neonates (LGA) to increased adiposity and T2DM also apply to diabetes exposure. The independent effect of in utero exposure to diabetes on these risks has been described as ‘glycemic memory’ [63].

The concept of glycemic memory was first established in adults and animal models exposed to hyperglycemia. Glycemic memory refers to the phenomenon of early glycemic load influencing later health [64]. This phenomenon was identified in two clinical trials, Diabetes Control and Complications Trial (DCCT) and UK Prospective Diabetes Study (UKPDS), which targeted early glycemic control in type 1 and type 2 diabetic patients, respectively [65,66]. Both trials demonstrated that if tight glycemic control was implemented early after diabetes onset, risk of diabetic complications was reduced. More importantly, follow-up studies of these trials (Epidemiology of Diabetes Interventions and Complications and UKPDS survivor cohort) revealed that the benefits of early intensive control remained prevalent more than a decade later, even when paired with poor glycemic control [66,67]. This can also be interpreted to say that poor glycemic control in early diabetes onset increases the risk for complications even when later glycemic control is improved [67].

Glycemic memory has been demonstrated in experimental models as well. Zebrafish with hyperglycemia from streptozotocin-induced diabetes exhibit inhibited wound healing and fin regeneration [68]. Even after streptozotocin withdrawal wound healing and fin regeneration remained impaired. This indicates that even when new “daughter tissues” are regenerated under euglycemic conditions, there is a cellular memory of exposure that influences the function of future tissues. Additionally, Caramori et al. (2015) revealed a glycemic memory in skin fibroblasts isolated from monozygotic twin pairs discordant for type 1 diabetes [69]. When re-exposed to high glucose in vitro, skin fibroblasts from type 1 diabetic (T1D) twins exhibited increased gene expression dysregulation compared to the non-T1D twins, 3308 differentially expressed genes compared to 889 genes, respectively [69].

Interestingly, the concept of glycemic memory was then expanded to high glucose exposure during pregnancy. A sibship study, performed by Dabelea and colleagues (2000) in the Pima Indian population, which has a high incidence and prevalence of type 2 diabetes due to both genetic and environmental factors [70], compared the prevalence of type 2 diabetes and BMI in siblings born before and after their mother was diagnosed with type 2 diabetes. They found that siblings exposed to diabetes in utero had an increased risk of developing type 2 diabetes themselves and had higher BMIs compared to their non-diabetic exposed siblings. These results emphasize the effects of a diabetic intrauterine environment independent of genetic predisposition, which is shared between siblings, on the development of metabolic disease in young adulthood. This was also validated in a more recent study of Swedish men, where BMI was on average 0.94 kg/m^2^ greater in men born to mothers with diabetes when compared to their brothers born before the mother was diagnosed [71].

A study by Boney et al. (2005) also illustrates the increased risk for metabolic disease in LGA offspring from mothers with GDM and/or obesity. With the rate of type 2 diabetes and obesity diagnoses increasing in childhood and adolescence [72], the authors sought to determine the contribution of GDM and LGA to the development of metabolic syndrome in children 6-11 year of age [73]. For these children, the definition of metabolic syndrome was defined as having two or more abnormal values from the following measurements: blood pressure, plasma insulin, glucose, triglyceride and HDL cholesterol. Children born LGA from GDM mothers had 50% prevalence of metabolic syndrome compared to 21%, 29%, and 18% for appropriate for gestational age (AGA) children born from GDM mothers, LGA children born from control mother, and AGA children born from control mothers, respectively. Furthermore, a multivariate logistic regression analysis determined that the interaction of LGA and GDM was significantly associated with insulin resistance in the 11-year-old children. In this study, LGA neonatal birth can be considered a surrogate marker of poor glycemic control during pregnancy, as it is the LGA/GDM combined condition that translates to increased risk of metabolic syndrome that is not seen when fetal growth is averaged in a diabetic intrauterine environment [73]. This was also suggested in a 2007 study from Hillier et al., where childhood obesity risk was decreased in correlation with treated maternal hyperglycemia when compared to untreated mothers [74]. A number of additional studies in human and rodent models have added evidence to the increased risk of metabolic syndrome components from intrauterine exposure to diabetes. The study of T2D youth also found that intrauterine exposure to diabetes is independent of maternal obesity increased risk of T2D in children and young adults. Furthermore, risk was higher when maternal obesity and diabetes occurred together [75]. Adolescents from diabetic mothers had increased obesity, systolic blood pressure, as well as 2-hr glucose and insulin levels compared to non-diabetes exposed adolescents [76]. These are all important metabolic parameters that can contribute to the development of metabolic syndrome. Animal models mirror these metabolic abnormalities in later life. Offspring from diabetic rats exhibit increased glucose, insulin, and triglyceride levels due to glucose intolerance and insulin resistance, which are markers of T2DM [77]. Furthermore, as the offspring of diabetic rats age, not only do they become obese but they also have increased liver triglyceride production and VLDL levels [78].

## 6. Sexual Dimorphism in Glycemic Memory

Growing evidence suggests that males and females have different physiologies not only after puberty, but also well before. For example, the growth hormone-IGF axis exhibits sexual dimorphism even at birth [79] and some evidence suggests that female fetuses are more insulin resistant than males [80]. Additionally, sex-specific responses to a range of intrauterine exposures, including maternal hyperglycemia are becoming increasingly evident. Findings based on the OBEGEST cohort study indicate that the risk of child overweight at 5–7 years increases with GDM exposure, but only in the male population [80] with a linear association between maternal blood glucose levels and boys’ BMI in the GDM group. Interestingly, no such relationship in girls born from GDM mother was found, although girls had an increase in adiposity when born from mother with intermediate glucose intolerance. Other studies also found greater susceptibility in male offspring with, for example, a study from Lingwood et al. showing a correlation between maternal blood glucose level and male adiposity only [81] or, a study based on Spanish population that shows an increased number of macrosomia in male affected by GDM [82] compared to female counterparts. In contrast, a study of the HAPO cohort (Hyperglycaemia and Adverse Pregnancy Outcome) [83], found a stronger graded effect of maternal hyperglycaemia on obesity and adiposity in girls than in boys at seven years of age. Also, it has been shown that treatment of mild GDM had a greater impact on male’s birthweight percentile and neonatal fat mass than in female suggesting that female neonates are more insulin resistant [84]. These findings illustrate that boys and girls differ in the immediate and latent responses to maternal hyperglycaemia in terms of adiposity and glucose metabolism. This discrepancy in findings is also likely due in part to differences in criteria used to define GDM as well as factors used to evaluate the metabolic impact in the neonate.

Rodent models designed to characterize the sexual dimorphism of intrauterine glycemic memory are also discordant. On one hand, studies have found greater dysfunction in glucose homeostasis, adiposity and cardiac health in females than in males mice exposed to high sucrose maternal diet during pregnancy [85,86]. Similarly, a maternal high fructose diet was shown to cause a decrease in placental weight, along with an increase in plasma leptin, fructose and glucose levels in female offspring, but not in male siblings [87]. In contrast, streptozotocin-based models and a genetic-based model (Akita mouse) of fetal hyperglycaemia results in male but not in female offspring hypertension [88,89] and metabolic dysfunction [90], respectively. Interestingly, in this latter publication the effect of hyperglycaemia was not limited to maternal hyperglycaemia but also found in the setting of paternal hyperglycaemia. The metabolic changes in offspring of paternal diabetes were milder than the effects of maternal diabetes. A possible explanation for the different effect-size might be that maternal hyperglycaemia results in direct programming of metabolic traits in the developing fetus, whereas the influence of paternal diabetes on offspring is indirect, possibly through transgenerationally transmitted epigenetics marks of fetal programming from the father himself.

Differences in the impact of early high-glucose exposure between male and female rodents can be partially explained by the sexual dimorphism observed in glucose metabolism related pathways. As discussed previously, oxidative stress plays a key role in subsequent glycemic memory. In both human [91] and rodent models [92], oxidative stress was higher in males compared to females. Similarly, ROS production also appears to be higher in males than in females [93] and clinical and experimental data suggest that females have a greater antioxidant potential than males [94]. Greater antioxidant potential seems to be in part due to estrogen. Interestingly, sexual dimorphism also affects the GSH synthesis pathway. In the liver, protein levels of glutamate cysteine ligase (GCL), an enzyme involved in GSH metabolism, are downregulated with age and correlated with a decrease in GSH content [95]. This decrease in GSH content was more drastic in male mice compared to females, as female mice demonstrated a higher GCL activity. During lung development, female mice show higher GPx1 transcription levels compared to males, suggesting that they may be better protected against oxidative stress [96]. Indeed, GPx1 is involved in the reduction of hydrogen peroxide into water. Cord blood samples from female neonates showed higher glutathione transferases (GST) activity than males [97]. GST play key roles in cellular detoxification against oxidative stress. Finally, when endothelial cells are challenged with an oxidative stimulus, female-derived tissues showed a two-fold increase of glutathione reductase (GR) activity, whereas no changes were observed in male-derived tissues [98]. GR is responsible for the GSH/GSSG ratio and thus essential for controlling the redox homeostasis. All together, these findings suggest that indeed glucose metabolism related oxidative stress could in part explain the sexual dimorphism observed in response to hyperglycemia. However, they cannot further explain the contradictory observations between male and female, as they seem to all agree in female being less sensitive to oxidative stress than male.

## 7. Epigenetic Glycemic Memory

The previously described studies illustrate that glycemic memory can last decades after the exposure occurs, therefore implicating epigenetic mechanisms as being involved. Inheritance of glycemic memory by daughter cells was exemplified by zebrafish hyperglycemia-exposed tissues, which demonstrated DNA hypomethylation persistence in daughter tissues that had never experienced hyperglycemia [68]. In contrast of metabolic memory, which depicts how nutrient access affects metabolic traits by epigenetic modifications, the literature in glycemic memory displays a sizable knowledge gap, with limited experimental data demonstrating that the increased risk of chronic diseases in the offspring of hyperglycemic mothers is associated to epigenetic mechanisms. Existing studies are almost exclusively focusing on two key developmental tissues: the placenta and the cord-blood tissues. These studies have been discussed elsewhere [99]. Here, we want to highlight key findings and provide with some perspective on mechanisms involved.

El Hajj et al. (2013) performed one of the first candidate gene studies to measure DNA methylation in neonates of mothers with GDM compared to euglycemic (non-GDM) mothers in both placenta and cord blood [100]. Of the 14 candidate genes tested, five genes (*MEST, IL10, NR3C1, OCT4, NDUFB6*) showed significant methylation changes between neonates of GDM mothers and non-GDM mothers. Similar study was then conducted using a genome wide approach in a total of 1,030 placental samples and confirm the epigenetic influence of GDM with evidence for placental global hypermethylation [101]. Following studies have highlighted several genes as sensitive to DNA methylation changes after GDM exposure [102,103,104,105]. Interestingly, among them LEP, ADIPOQ and MEST were also found differentially methylated in blood from obese adults [100,106] confirming the influence of epigenetic modifications in long-term consequences of early exposure to hyperglycemia. Epigenetic changes have also been found at the level of histones. In humans, evaluation of histone modifications in DCCT/EDIC study subjects found an increased number of genomic regions occupied by H3K9ac, a marker of active transcription [107]. On the other hand, transient in vitro high-glucose exposure of aortic endothelial cells stimulated H3K4me1 enrichment at the promoter of the NF-κB p65 subunit, likely mediated by Set7 methyltransferase [108]. These changes in H3K4me1 remained for at least 6 days in normoglycemic conditions. High-glucose was also shown to induce NF-κB activation in THP-1 monocytes with associated increases in NF-κB target genes, such as the pro-inflammatory TNF-α and IL-6 [109]. Furthermore, the increased expression of these genes was associated with increased recruitment of the p300 histone acetyltransferase (HAT).

Growing evidence proposes that nutrients play a crucial role in the regulation of epigenetic modifiers through nutrient pathways associated cofactors or oxidative stress, providing with possible mechanism reconciliating nutrient availability, metabolite levels and epigenetic regulation. Indeed, various nutrients and their metabolites function as substrates or cofactors for epigenetic modifiers, thus, nutrition would be able to modulate or reverse epigenetics marks. Interestingly, the glucose associated nutrient sensing pathways are releasing key epigenetic modifiers such as O-GlcNAc, Acetyl-CoA, and NAD+ (Figure 1).

Exposure to excess nutrient during pregnancy (GDM) has been shown to result in ROS increase in exposed neonates [110] and evidence supports the influence of ROS on epigenetic mechanisms. Indeed, oxidant molecules can directly interact with DNA causing both genetic, as well as, epigenetic changes [111]. ROS can lead to the formation of 5hmC from 5mC [112] and affect DNA methylation via oxidation of guanosine to 8-oxo-2′-deoxyguanosine [113]. Similarly, ROS can directly modify histones [114]. Alternatively, ROS can have an indirect effect on methylation by reducing S-Adenosyl methionine (SAM) availability [114].

Increasing the O-GlcNAc levels in a mouse β-cell line, through high glucose or pharmacologic treatment, induced increased enrichment of H3K4me3, H3K9ac, and histone GlcNAc at the *Ins2* promoter [115]. Generally, O-GlcNAcylation is described as a post-translational modification in histones and is also involved in DNA methylation [116]. O-GlcNAcylation is directly dependent on the metabolite state of the cell, since intracellular UDP-GlcNAc is synthetized through the hexosamine biosynthesis pathway, which is dependent on glucose availability. LSD2/KDM1B, a member of the LSD family that catalyzes demethylation of H3K4Me1 and H3K4me2, has recently been shown to have E3 ubiquitin ligase activity to promote the proteasome-mediated degradation of O-GlcNAc transferase (OGT), an enzyme required to add the sugar moiety O-GlcNAc as a posttranslational modification to a variety of substrates [117]. Finally, in mouse embryonic stem cells, OGT regulates TET1 activity at CpG-rich sequences near gene promoters [118]. Furthermore, TET-OGT interactions promote O-GlcNAcylation of host cell factor 1 (HCF1), a component of the H3K4 methyltransferase SET1/COMPASS complex, to stimulate trimethylation of H3K4 at gene promoters [119]. Thus, DNA and histone methylation can be regulated through TET-OGT-mediated epigenetic changes in response to nutrient availability.

Acetyl-CoA is a universal substrate for the acetylation of histones. The activity of HATs relies on intracellular levels of acetyl-CoA, then representing a prominent example of the interplay between metabolism and chromatin dynamics. Several studies support the direct interconnection between histone acetylation and availability of acetyl-CoA. For example, yeast grown in conditions of high acetyl-CoA precursors such as glucose, exhibit histone hyperacetylation [120]. Studies in mammalian cells also support a role for acetyl-CoA in transcriptional activation via histone acetylation in response to glucose availability [121].

In metabolism, NAD functions as an electron transfer molecule in redox reactions, and it is found in the form of an oxidizing (NAD+) or a reducing agent (NADH). NAD+ participates in various oxidative pathways such as glycolysis as previously discussed. NAD+ is an obligated cofactor for the class III histone deacetylase (HDAC) enzymes known as sirtuins (SIRT), thus, the activity of sirtuins is dependent on nutrient availability. Additionally, it has been shown that KDM1A/LSD1 forms a corepressor complex with the NAD+ dependent HDAC SIRT1 to triggers the deacetylation of H4K16ac and demethylation of H3K4me in human cell lines [122]. Collectively, the sirtuin-mediated histone deacetylation and its dependency on NAD+ levels establish an intimate relationship between chromatin dynamics and metabolism. This relationship is also highlighted by the influence of resveratrol, a known activator of SIRT1, on metabolic fetal programming. Indeed, resveratrol treatment of rat GDM dams results in improvement of obesity-related complications, glucose homeostasis and islet function in young offspring [123]. Altogether, these observations provide strong evidence to support glucose exposure and glucose sensing pathways as key mechanisms involved in long-term consequences of early hyperglycemia through epigenetic glycemic memory.

## 8. Conclusions

Strong evidence demonstrates that offspring exposed to a detrimental intrauterine environment related to maternal nutritional status are programmed to develop a number of chronic diseases such as obesity and diabetes, perpetuating the vicious circle of metabolic diseases across generations (Figure 3).

Although there are gaps in our knowledge of the mechanisms involved, in this review, we described a comprehensive view of the relationship between glucotoxicity, oxidative stress and glycemic memory as one of the potential mechanisms to support the observations of long-term consequences of early high glucose exposure in metabolic disease susceptibility. Importantly, further exploration is needed to include multiple levels of “omic” (transcriptomic, epigenetic, metabolomic), translational and longitudinal studies to gain insight into the sexual dimorphic and transgenerational aspects of prenatal hyperglycaemia associated with fetal programming. Indeed, more direct evidence of transgenerational programming in humans is still needed and will be key to fully measuring the impact of maternal hyperglycaemia on the long-term health of the next generation.

## Figures and Tables

**Figure 1 biomolecules-10-01359-f001:**
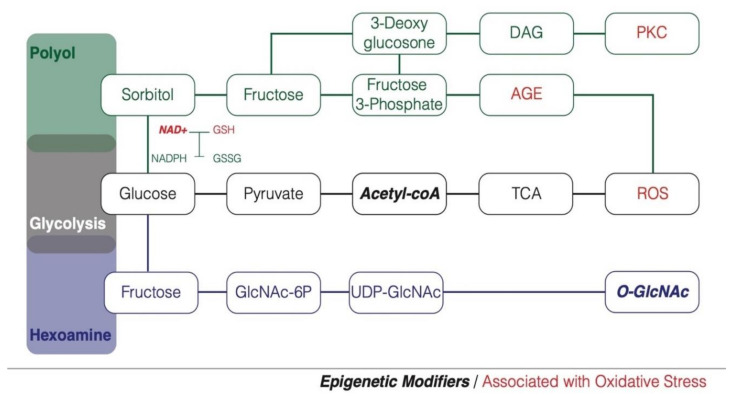
Pathways involved in Glucose metabolism with influences on oxidative stress. The Polyol and Hexoamine pathways are recruited when glucose rises to compensate the Glycolysis pathway. The activation of Polyol and Hexoamine pathways participate to the cellular glucotoxicity by increasing the production of oxidative stress associated molecules (in red). The increase of glucose metabolism also leads to an increase of epigenetic modifiers (in bold italic). Such factors may explain the association between high-glucose exposure and increased metabolic sensitivity. DAG: diacylglycerol; PKC: protein kinase C; AGE: advanced glycation end-products; NAD: nicotinamide adenine dinucleotide; NADPH: nicotinamide adenine dinucleotide phosphate; GSH: glutathione; GSSG: glutathione disulfide; TCA: tricarboxylic acid cycle; ROS: reactive oxygen species; GlcNAc-6P: N-acetylglucosamine-6-phosphate; O-GlcNAc: O-Linked β-N-acetylglucosamine.

**Figure 2 biomolecules-10-01359-f002:**
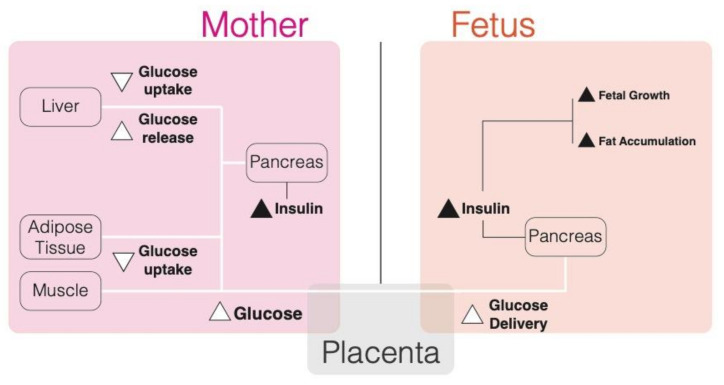
Glucose metabolism during pregnancy and associated consequences. White lines represent glucose transfer while black lines represent insulin secretion.

**Figure 3 biomolecules-10-01359-f003:**
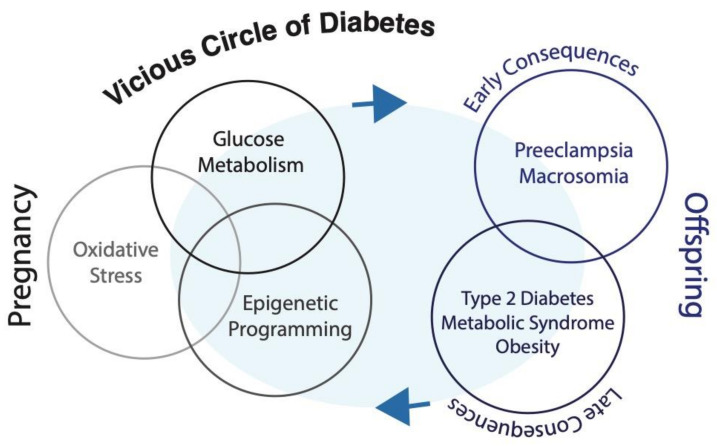
The Vicious Circle of Diabetes.

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
