# Peer review of "Prenatal Hyperglycemia Exposure and Cellular Stress, a Sugar-Coated View of Early Programming of Metabolic Diseases"

_biomolecules, 2020, doi:10.3390/biom10101359_

Round 1

Reviewer 1 Report

Tozour et al. nicely summarized and provided recent research progresses on the role of maternal hyperglycemia in offspring’s’ metabolic outcomes.

They also presented well-collected references on this topic. Thus, I think the authors’ paper fits well the interest of this journal’s audience.

There are a couple of things I would like to comment on, which I wish the authors can address if possible. First, the authors wrote hyperglycemia causes cellular stresses like ROS and also maternal hyperglycemia-induced offspring’s epigenetic changes. However, I do not see any detailed/mechanistic explanation how cellular stresses like ROS influence offspring’s epigenetic changes and metabolic outcomes. I think ROS part is somewhat a separated topic in this manuscript.

Also, maternal hyperglycemia during lactation may potentially influence offspring’s metabolic outcomes as described by Lippert et al (2020)(1). Thus, the authors may explain such possibilities beside the epigenetic changes induced from intrauterine environment.

  1. R. N. Lippert et al., Maternal high-fat diet during lactation reprograms the dopaminergic circuitry in mice. J Clin Invest 130, 3761-3776 (2020).

Author Response

We thank the reviewers for the substantial amount of time they have invested in reviewing this manuscript, and for their insightful comments. We have attempted to address their comments in the manuscript. The specific responses to the reviewers’ comments are described below.

Tozour et al. nicely summarized and provided recent research progresses on the role of maternal hyperglycemia in offspring’s’ metabolic outcomes.

They also presented well-collected references on this topic. Thus, I think the authors’ paper fits well the interest of this journal’s audience.

There are a couple of things I would like to comment on, which I wish the authors can address if possible. First, the authors wrote hyperglycemia causes cellular stresses like ROS and also maternal hyperglycemia-induced offspring’s epigenetic changes. However, I do not see any detailed/mechanistic explanation how cellular stresses like ROS influence offspring’s epigenetic changes and metabolic outcomes. I think ROS part is somewhat a separated topic in this manuscript.

We thank the reviewer for his comment. We believe that the reviewer was referring to chapter 7 on Epigenetic Glycemic Memory. We agree with the reviewer that, as previously written, the connection between ROS and epigenetic modifications was not strong enough. Indeed, the previous version focused on nutrient-related epigenetic modifiers that are counterparts of the ROS when considering glucose metabolism pathways. We have now rewritten this part, added a paragraph (lines 406-412) dedicated to ROS epigenetic influences, and included new references (110-114). The aim of this chapter is 1) to highlight evidence of epigenetic alterations in the context of early hyperglycemia and 2) to identify possible mechanisms responsible for the epigenetic glycemic memory related to glucose metabolism. As ROS production is inherent to glucose metabolism, it appears to us key to mention it as a possible player. We also have partially rewritten lines 398 to 404 to moderate our statement. 

Also, maternal hyperglycemia during lactation may potentially influence offspring’s metabolic outcomes as described by Lippert et al (2020)(1). Thus, the authors may explain such possibilities beside the epigenetic changes induced from intrauterine environment.

The reviewer is correct that epigenetic changes and reprogramming are not limited to intrauterine development and that lactation is part of the critical window for perinatal development. It has been previously shown that long-term consequences will vary depending on the time of exposure but also, on the type of exposure emphasizing the need to consider both criteria when studying fetal programming as mentioned by the reviewer. However, the goal of this manuscript was not to exhaustively review fetal programming, but rather, to focus on perinatal hyperglycemia and associated glycemic memory, an aspect that we believe has been so far understudied. We have partially rewritten the conclusion (lines 457-465) to clarify this point. 

Reviewer 2 Report

It is a well-written review to summarize the connection between maternal, even paternal metabolic derangement and offspring metabolic syndromes. It has the clinical aspects and also molecular explanations, which is excellent for a general audience.

Minor,

  1. For Figure-1, it will be best to provide a concise explanation of the figure (I know that you explained it in the text) and provide a list of abbreviations so I the reader doesn’t need to find them in the text.
  2. For 4.1. Glucose Metabolism during Pregnancy. Please specify the insulin resistance is for the mother. It is a little confusing whether these statements are specific for the mother or the fetus or both. Based on my reading, it is only for the mother.
  3. There is a recent publication in Nature showing that regulation of NADH/NAD+ ratio in the liver has an effect on insulin resistance (1), which also reflects GCKR polymorphism. I am wondering whether there are any data of gene polymorphism of GCKR or genes regulating NADH/NAD+ in the maternal – offspring “glycemic memory” studies.
  4. The relationship of NADH/NAD+ with sirtuins is very interesting. Has anybody ever tried to use sirtuin activators such as resveratrol to abolish the maternal – offspring “glycemic memory” in animals?

Reference

1. Goodman RP, Markhard AL, Shah H, Sharma R, Skinner OS, Clish CB, Deik A, Patgiri A, Hsu YH, Masia R, Noh HL, Suk S, Goldberger O, Hirschhorn JN, Yellen G, Kim JK, Mootha VK. Hepatic NADH reductive stress underlies common variation in metabolic traits. Nature. 2020;583(7814):122-6. Epub 2020/05/29. doi: 10.1038/s41586-020-2337-2. PubMed PMID: 32461692.

Author Response

We thank the reviewers for the substantial amount of time they have invested in reviewing this manuscript, and for their insightful comments. We have attempted to address their comments in the manuscript. The specific responses to the reviewers’ comments are described below.

It is a well-written review to summarize the connection between maternal, even paternal metabolic derangement and offspring metabolic syndromes. It has the clinical aspects and also molecular explanations, which is excellent for a general audience.

For Figure-1, it will be best to provide a concise explanation of the figure (I know that you explained it in the text) and provide a list of abbreviations so I the reader doesn’t need to find them in the text.

We have modified the manuscript accordingly (lines 62-71).

For 4.1. Glucose Metabolism during Pregnancy. Please specify the insulin resistance is for the mother. It is a little confusing whether these statements are specific for the mother or the fetus or both. Based on my reading, it is only for the mother.

We appreciate the reviewer's comment and apologize for any confusion. To clarify, we have now added maternal to the title: «4.1 Maternal Glucose Metabolism during Pregnancy » (line 147).

There is a recent publication in Nature showing that regulation of NADH/NAD+ ratio in the liver has an effect on insulin resistance (1), which also reflects GCKR polymorphism. I am wondering whether there are any data of gene polymorphism of GCKR or genes regulating NADH/NAD+ in the maternal – offspring “glycemic memory” studies.

We thank the reviewer for this comment. Several studies have indeed identified rs780094, a polymorphism associated with GCKR, as a risk factor for gestational diabetes mellitus (Gao et al. ,2016; Anghebem et al., 2017; Jamalpour et al., 2018). Thus, supporting the association between NADH/NAD+, glucose metabolism, and insulin resistance. However, if such finding highlights possible mechanism associated with insulin resistance in GDM, we believe it does not provide with further evidence on the possible role played by NAD+ in the context of glycemic memory associated consequence and thus, fall outside of the scope of our review.

The relationship of NADH/NAD+ with sirtuins is very interesting. Has anybody ever tried to use sirtuin activators such as resveratrol to abolish the maternal – offspring “glycemic memory” in animals?

Again, we thank the reviewer for his interesting comment on the association between resveratrol and glycemic memory, an aspect that was not addressed in the submitted manuscript. Several studies have been performed to better understand the impact of resveratrol exposure during pregnancy and associated fetal outcomes in various conditions (for review Darby et al. 2019). One publication from Brawerman et al. (2019) has focused on gestational diabetes and has shown improvement of the fetal outcomes in young rats either under low or high-fat diet after resveratrol treatment. These findings are very interesting as they further support the association between NAD+, SIRT, and long-term outcomes of early hyperglycemia exposure. We have now incorporated this study in the manuscript (lines 443-446).

References:

Anghebem-Oliveira MI, Webber S, Alberton D, et al. The GCKR Gene Polymorphism rs780094 is a Risk Factor for Gestational Diabetes in a Brazilian Population. J Clin Lab Anal. Mar 2017;31(2)doi:10.1002/jcla.22035

Brawerman GM, Kereliuk SM, Brar N, et al. Maternal resveratrol administration protects against gestational diabetes-induced glucose intolerance and islet dysfunction in the rat offspring. J Physiol. Aug 2019;597(16):4175-4192. doi:10.1113/JP278082

Darby JRT, Mohd Dollah MHB, Regnault TRH, Williams MT, Morrison JL. Systematic review: Impact of resveratrol exposure during pregnancy on maternal and fetal outcomes in animal models of human pregnancy complications-Are we ready for the clinic? Pharmacol Res. Jun 2019;144:264-278. doi:10.1016/j.phrs.2019.04.020

Gao K, Wang J, Li L, et al. Polymorphisms in Four Genes (KCNQ1 rs151290, KLF14 rs972283, GCKR rs780094 and MTNR1B rs10830963) and Their Correlation with Type 2 Diabetes Mellitus in Han Chinese in Henan Province, China. Int J Environ Res Public Health. Feb 26 2016;13(3)doi:10.3390/ijerph13030260

Jamalpour S, Zain SM, Mosavat M, Mohamed Z, Omar SZ. A case-control study and meta-analysis confirm glucokinase regulatory gene rs780094 is a risk factor for gestational diabetes mellitus. Gene. Apr 15 2018;650:34-40. doi:10.1016/j.gene.2018.01.091